# T Lymphocyte and CAR-T Cell-Derived Extracellular Vesicles and Their Applications in Cancer Therapy

**DOI:** 10.3390/cells11050790

**Published:** 2022-02-24

**Authors:** Victor Calvo, Manuel Izquierdo

**Affiliations:** 1Departamento de Bioquímica, Instituto de Investigaciones Biomédicas Alberto Sols CSIC-UAM, Facultad de Medicina, Universidad Autónoma de Madrid, 28029 Madrid, Spain; vcalvo@iib.uam.es; 2Departamento de Metabolismo y Señalización Celular, Instituto de Investigaciones Biomédicas Alberto Sols, CSIC-UAM, 28029 Madrid, Spain

**Keywords:** exosomes, T lymphocytes, immune synapse, secretory granules, multivesicular bodies, cytotoxic activity, cell death, CAR T lymphocytes

## Abstract

Extracellular vesicles (EV) are a very diverse group of cell-derived vesicles released by almost all kind of living cells. EV are involved in intercellular exchange, both nearby and systemically, since they induce signals and transmit their cargo (proteins, lipids, miRNAs) to other cells, which subsequently trigger a wide variety of biological responses in the target cells. However, cell surface receptor-induced EV release is limited to cells from the immune system, including T lymphocytes. T cell receptor activation of T lymphocytes induces secretion of EV containing T cell receptors for antigen and several bioactive molecules, including proapoptotic proteins. These EV are specific for antigen-bearing cells, which make them ideal candidates for a cell-free, EV-dependent cancer therapy. In this review we examine the generation of EV by T lymphocytes and CAR-T cells and some potential therapeutic approaches of these EV.

## 1. Introduction

### Extracellular Vesicle Types

Extracellular vesicles (EV) are a very diverse group of cell-derived, lipid bilayer enclosed vesicles released by almost all kind of living cells, and this process has been highly conserved throughout evolution [1]. There are several subtypes of EV that have distinctive structural, biochemical properties and composition depending of their intracellular origin that, in turn, affect their function [2]. EV are highly heterogeneous, which in great part is responsible for hindering the characterization and description of their properties and functions [1], and are often classified in terms of their generation mechanism. The first type includes the EV released by dying apoptotic cells, which are called apoptotic bodies. Apoptotic bodies have a wide range of sizes and exhibit different compositions and features from EV derived from living cells, and are not discussed in this review. The second type of EV are directly formed by outward budding of the plasma membrane and are called microvesicles, ectosomes, ectovesicles or, generically, shedding vesicles [3] (Figure 1). The third type, exosomes, are secreted via the fusion of multivesicular bodies (MVB) with the plasma membrane [4,5] and have diameters ranging 30–150 nm (depending of the estimation technique [6]), smaller than shedding vesicles (100 nm to >1 μm) (Figure 1). MVB are subcellular organelles containing intralumenal vesicles (ILV) [7,8] that are components of the endolysosomal system, which also comprises early endosomes, late endosomes and lysosomes [9,10]. MVB are formed by inward budding from the external membrane of late endosomes and successive pinching off of budding vesicles into the lumenal space of late endosomes (Figure 1). ILV present in MVB are called exosomes when they are released into the extracellular medium [11].

The term exosomes (initially used to define shedding vesicles with diameters ranging from 40 to 1000 nm [45]) was later adapted to define nanovesicles of endosomal source that are liberated by fusion of MVB with the plasma membrane, as a means to discard specific out of date constituents during red cell maturation [4], and since then this proposal has been widely accepted by the scientific community, although not fully standardized yet. However, growing evidence supports that EV in general, and exosomes in particular, have much wider biological functions than removal of certain unwanted proteins, and all these EV are involved in intercellular communication, both locally and systemically, since they may transfer their cargo (proteins, lipids, miRNAs) between cells, and also may trigger new cues in recipient or target cells [1,10,12,46]. Thus, EV have been shown to affect the physiology of neighbouring target cells in diverse ways, from inducing cell signaling upon cell surface receptor triggering, to generating new properties in the target cells after acquisition of novel receptors, enzymes or genetic material contained into the EV [2,47].

## 2. Exosomes and Extracellular Vesicles: Normalization Attempts and Isolation Protocols

Exosomes are just one type of EV. The International Society of Extracellular Vesicles (ISEV, the major scientific society on EV research, https://www.isev.org/, accessed on 24 January 2022) recommends “extracellular vesicles” as the “generic term for particles naturally released from the cell that are delimited by a lipid bilayer and cannot replicate, i.e., do not contain a functional nucleus” [48] including ectosomes, microvesicles, microparticles (different types of plasma membrane-shedding vesicles), apoptotic bodies and exosomes (endosomal origin). Regarding EV characterization, the “minimal experimental requirements for definition of extracellular vesicles” from ISEV or “MISEV” stipulate recommendations on experimental methodology and minimal information on reporting EV isolation/purification, EV description and studies on EV biological function [49,50]. In this context, although some molecules have been proposed as specific EV markers [48,49], they do not allow distinguishing among the different EV types [48,49,51]. In conclusion, when an isolated EV preparation is described, the more general term EV is recommended unless the MVB origin of the vesicles it contains has been unambiguously established [13]. We keep the ISEV endorsement of the inclusive term EV along this review unless the classification of shedding vesicle or exosomes is evident using ISEV criteria.

The isolated material obtained from several EV sources (biological fluids and cell culture supernatants) generally contains variable proportions of diverse EV, and the composition of the recovered EV may vary vastly depending of the protocol [13,49]. Therefore, as the first consideration, given the heterogeneity within the preparations, it is problematic to assign a biological effect to any EV contained in the preparations. Accordingly, “MISEV” recommendations include biochemical (i.e., protein markers), biophysical (i.e., single vesicle characterization by electron microscopy (EM) and nanoparticle tracking analysis (NTA)) and functional tools (i.e., EV depletion to remove biological activity) that may allow researchers to ascribe any potential function to EV.

As the second issue, although numerous analyses have highlighted proteins, miRNAs and lipids commonly found in exosomes and EV preparations, even curated by ISEV standards (see VESICLEPEDIA: http://www.microvesicles.org/, accessed on 24 January 2022 and EXOCARTA: http://exocarta.org/index.html#, accessed on 24 January 2022) it is clear that these studies do not denote “exosome-specific” markers or “EV-specific” markers, but rather “exosome-enriched” (or “EV-enriched”) proteins. For instance, CD63 and CD81, which are reported to be enriched in exosomes produced by a wide variety of cell types studied (EXOCARTA: exosomal markers) and have been considered as canonical exosome markers [13], are also present at the plasma membrane of T and B lymphocytes, most probably as a consequence of the diffusion of the CD63 located at the limiting membrane of MVB after their constitutive fusion with the plasma membrane [18,52] and hence located in shedding vesicles (and, quite probably, in apoptotic bodies) (Figure 1). Considering this caveat, the EV field has proposed a list of EV-specific markers [48,49], but has not distinguished among EV subtypes [51], such as those produced either via shedding from the plasma membrane or via endosomal compartments [48,49]. In this context, although MISEV 2018 recommendations do not define a universal negative marker for any given EV source, markers should be chosen to reveal the level of “contaminants” in that specific EV material [48,53]. This may be applied to different EV subtypes by establishing specific markers of their subcellular origin that are reliable within the experimental system [48]. Accordingly, some molecules can be used as negative markers for certain EV types and particular cell types. As an example, the evaluation of both positive exosome markers (CD63 and lysosomal-associated membrane protein 1 -LAMP-1-) and negative markers (i.e., CD45 and CD28, which are very abundant in the T lymphocyte plasma membrane and hence in plasma-membrane derived EV and apoptotic bodies, but absent in exosomes [54]) in EV preparations isolated by differential centrifugation allowed characterization of the isolated EV as canonic exosomes [19]. A similar approach was followed by other researchers in order to characterize as bona fide exosomes the EV produced by chimeric antigen receptor (CAR) T lymphocytes in a preclinical in vivo model [55]. The CD63^+^, LAMP-1^+^, CD45^−^ and CD28^−^ T lymphocyte-derived exosomes were subsequently confirmed in their endosomal/MVB origin by analyses of CD63^+^ MVB fusion with the plasma membrane in living T cells [18].

In conclusion, EV samples most probably contain a mixture of EV of exosomal and non-exosomal types; therefore, unless their MVB origin has been unambiguously established, it is recommended to use the more general name EV [13]. We follow this criterion in this review.

## 3. Exosome Biogenesis, Composition and Regulation of Exosome Secretion

Several researchers have demonstrated the endosomal origin of exosomes by using EM [10,56] or fluorescence living-cell imaging [18]. Solid evidence of their endosomal origin is often difficult to obtain, since MVB fusion with the plasma membrane is a very dynamic and stochastic event [52] that is difficult to document using EM. Despite this caveat, several researchers have obtained definitive data regarding their MVB origin by using EM [10,56] or fluorescence living-cell imaging [18,52]. ILV are formed by inward budding of the membrane of late endosomes by at least three distinct molecular mechanisms that have been partially characterized (Figure 1). It is out of the scope of this review to extensively explain these mechanisms (please refer to excellent published reviews on this topic [1,6,22,57]). Therefore, we first discuss MVB and exosome generation mechanisms in general terms, and subsequently we focus on the mechanisms operating in T lymphocytes.

The first mechanism controlling ILV generation involves ESCRT members [7]. However, exosomes can also be created in an ESCRT-independent manner since MVB containing ILV filled with CD63 are still formed upon elimination of the components of the four ESCRT complexes (reviewed in [1]). The second mechanism involves the action of certain lipids such as ceramide [22,58,59] and DAG [17,18]. This first ESCRT-independent mechanism of exosome biogenesis requires the production of ceramide by nSMase2, which hydrolyses sphingomyelin to ceramide [59]. Ceramide, a cone-shaped lipid, then allows the generation of membrane subdomains, which impose a spontaneous negative curvature on the limiting membrane of late endosomes favoring inward budding of ILV (Figure 1). Moreover, purified exosomes were enriched in ceramide, and the release of exosomes was decreased after the pharmacologic inhibition of nSMase or nSMase interference [59]. Vice versa, inhibitors of sphingomyelin synthase (i.e., D609) that enhance ceramide levels have been shown to increase exosome secretion [60]. Inverted cone-shaped lipids such as LBPA are also abundant in internal membranes (ILV) of MVB [61]. LBPA controls the formation of ILV both in vitro and in vivo via the recruitment of ESCRT-family member Alix [62]. Thus, interconnection between ESCRT-dependent and ESCRT-independent lipid-mediated pathways may exist (Figure 1) [6]. In addition, phospholipase D2 (PLD2), activated by ARF6 small GTPase, causes the hydrolysis of phosphatidylcholine to phosphatidic acid (PA), a cone-shaped lipid as ceramide, which has been shown to be necessary for ILV biogenesis and exosome secretion [63,64]. The model proposed is that PA formation in the inner leaflet of the MVB limiting membrane induces inward curvature, and thus formation, of ILV [63], as described for ceramide [59]. In addition, cholesterol confers high fluidity to lipid bilayers and is required for the formation of highly curved membrane structures such as caveolae and synaptic vesicles. In this context, caveolin-1 has been recently shown to act as a cholesterol rheostat in MVB, regulating ILV biogenesis and exosomal protein cargo sorting through the control of cholesterol content at the endosomal compartment/MVB and hence regulating exosome secretion [65].

The third mechanism for exosome biogenesis involves proteins of the tetraspanin (TSP) family that have been shown to regulate ESCRT-independent endosomal sorting (reviewed in [1,6,22]. Tetraspanins such as CD63, CD9 and CD81 form clusters with other tetraspanins and with diverse transmembrane proteins and certain lipids in membrane domains known as tetraspanin-enriched microdomains. Apparently, tetraspanins facilitate the formation of the membrane microdomains that undergo inward budding to facilitate ILV formation [1,22]. For instance, CD81 has a cone-shaped structure with an intramembrane cavity that accommodates cholesterol and that is likely to be common to other tetraspanins. Clustering of these cone-shaped tetraspanins/cholesterol complexes could then induce the inward budding of the microdomains in which they are enriched [1], which constitutes an example of crosstalk between tetraspanin and lipid-mediated mechanisms involved in ILV biogenesis [6]. In addition, CD63 may also trigger ESCRT-independent and ESCRT-dependent endosomal sorting, which is another example of crosstalk during MVB biogenesis [66] (Figure 1). MAL is a membrane protein containing four transmembrane segments and its expression is restricted to T lymphocytes and polarized epithelial and myelin-forming cells. MAL appears also to be responsible for ILV biogenesis and exosome secretion in T lymphocytes, since MAL silencing in T lymphocytes reduces constitutive exosome secretion and generates immature, aberrant MVB that do not fuse with the plasma membrane and merge with lysosomes [22,23] (Figure 1). Another pathway that regulates MVB traffic towards lysosomes for degradation is ISGylation (Figure 1). Interferon-stimulated gene 15 (ISG15) is a ubiquitin-like protein, and ISG15-dependent ISGylation of ESCRT family member TSG101 protein provokes its aggregation and degradation, being sufficient to decrease constitutive exosome secretion in T lymphocytes. Although inducible, TCR-controlled exosome secretion was not evaluated in these experiments [24].

Apart from ILV biogenesis, the transport, docking and fusion of mature MVB with the plasma membrane are crucial steps implicated in the secretory traffic of exosomes, which are strictly regulated. It is out of the scope of this review to deal in depth with all the pathways, molecular components and mechanisms involved in MVB polarized traffic, so please refer to recent and excellent reviews on this subject for details [1,6,13]. MVB may fuse either with lysosomes (Figure 1) for degradation of their cargo, or with the plasma membrane. In both cases, three steps (transport, docking and fusion) are required, but the effectors involved in directing MVB to lysosomes or the plasma membrane are distinctive. In general, MVB intracellular transport comprises the association of these organelles with the cytoskeleton (actin filaments and microtubules), associated molecular motors (dynein, kinesin, among others) and some molecular switches (Rab GTPases) [1]. In brief, there are three major group of regulators of MVB transport and fusion with the plasma membrane that include Rab GTPases, actin/tubulin cytoskeleton and soluble N-ethylmaleimide-sensitive fusion attachment protein receptors (SNAREs) [1,13]. Among these, Rab GTPase members control different steps of intracellular vesicular traffic, such as vesicle budding, vesicle and organelle mobility through cytoskeleton interaction and vesicle docking to target compartments, leading to membrane fusion, and play a very conserved role during MVB traffic to the plasma membrane in diverse cell types (reviewed in [6,64,67]). The strategies directed to unveil the role of Rab GTPases in exosome secretion are mainly based on overexpression of dominant negative mutants or silencing by using small hairpin RNA (shRNA) [6,64]. Thus, loss of function approaches to interfere with diverse Rab GTPases (Rab7, Rab11, Rab27a, Rab27b and Rab35) in different cell types have resulted in inhibition of MVB transport and exosome secretion, although particular Rabs may act in different cell types [6,64] (Figure 1). The microtubule network is needed for MVB transport to the plasma membrane for subsequent exosome release [13,67] (towards microtubule plus ends in the centrosome, at least in the context of the immune synapse-IS), although MVB can also be targeted towards lysosomes via retrograde transport on microtubules (towards microtubule minus ends) due to the retrograde molecular motor dynein (Figure 1) [1]. Regarding actin cytoskeleton, cortical F-actin depolymerization at the central IS is needed to allow MVB docking to the plasma membrane and exosome release [31] and, in T and B lymphocytes, the dismantling of F-actin pool around microtubule-organizing center (MTOC) and MVB is involved in MTOC/MVB polarization [32,68,69]. In the last step of exosome secretion, SNAREs (synaptosomal protein 23 -SNAP23-, VAMP7, VAMP8, YKT6, syntaxins) are required for calcium-dependent MVB fusion with the plasma membrane (reviewed in [1]). Thus, the molecular components involved in these stages have been partially identified and have been shown to be common, but may also differ, among different cell types [1,67].

All the mechanisms for MVB maturation also control, during ILV formation from the limiting endosomal membrane, that certain proteins, lipids, and nucleic acids are specifically sorted to ILV, and subsequently constitute exosome cargo (reviewed in [6]) (Figure 2). Please consult the web pages applying the ISEV standards concerning proteins, miRNAs and lipids frequently found and enriched in exosomes and other EV (EXOCARTA: http://exocarta.org/index.html#, accessed on 24 January 2022 and VESICLEPEDIA: http://www.microvesicles.org/, accessed on 24 January 2022). Exosome cargo reflects, in part, the composition of the donor cell but is also a consequence of a regulated sorting mechanism [1]. Exosomal cargo comprises several proteins including receptors, transcription factors, transmembrane proteins, enzymes, extracellular matrix proteins, but also lipids and nucleic acids (DNA, mRNA, and miRNA), inside and on the surface of exosomes [1] (Figure 2).

Analysis of exosome protein composition reveals that some proteins are common among exosomes from different cellular origins, whereas some proteins are specific of cells and tissues that secrete them [1,70]. Exosomal proteins common to different cell types include adhesion molecules such as cell adhesion molecules (CAM), integrins, tetraspanins (CD63, CD9, CD81), MHC-I, a range of membrane fusion and transferring proteins (Rab2, Rab5, Rab7, flotillin 1, annexins), heat shock proteins (HSP70, HSP90), cytosolic proteins involved in MVB formation (Alix, Tsg101) and cytosolic enzymes (GAPDH) [1,71,72] (see also EXOCARTA http://exocarta.org/, accessed on 24 January 2022) (Figure 2). MHC-II is specific of B lymphocytes and antigen-presenting cells (APC)-derived exosomes [73], and transferrin receptors (TfR) are specific of reticulocyte-derived exosomes [1]. BCR and TCR are specific to B or T lymphocyte-derived exosomes, respectively [54,74]. The Fas ligand (FasL) and Apo2 ligand (Apo2L) appear to be more restricted to T lymphocyte-derived exosomes [73,75] and exosomes from certain tumor cells [76]. Perforin and granzyme A and B have been shown to be associated with exosomes from cytotoxic T lymphocytes (CTL) and natural killer cells (NK) [77,78]. It is not clear whether perforin/granzymes reside inside ILV or, alternatively, are externally associated to ILV membrane [77,79] (Figure 2). However, it has been shown by exosome surface labelling and flow cytometry [80], that the later situation may occur in chimeric antigen receptor (CAR) T-cell derived exosomes, facilitating perforin-granzyme binding and activity on target cells [55,80]. If this is confirmed for different CAR T-cell and T-cell derived exosomes, it will be necessary to understand how these cytosolic proteins are sorted or translocated to the luminal side of the ILV. The fact these proapoptotic proteins are not integral membrane proteins would suggest that association with ILV limiting membrane, and hence with exosomes, could be mediated by unknown membrane adapter protein(s). Alternatively, if perforin and granzyme are located inside exosomes, further research is necessary to understand how these proteins are functionally delivered to target cells.

The lipid cargo of exosomes is both cell-specific and common. Lipids not only have a significant function in maintaining exosome shape and curvature but also participate in exosome biogenesis (see above). LBPA interaction with Alix facilitates the inward budding of the MVB limiting membrane [62], which constitutes an example of crosstalk between ESCRT and lipid-mediated mechanisms involved in ILV biogenesis (Figure 2). Sphingomyelin, phosphatidylcholine and LBPA help in discriminating numerous types of vesicles. Diverse types of microvesicles have a similar content of sphingomyelin and phosphatidylcholine, while sphingomyelin concentration is higher in exosomes. LBPA is exclusive to endosomes and exosomes [81]. During membrane rearrangements occurring during EV generation there are changes in the lipid asymmetry of membrane phospholipids (PS exposition from the inner to the outer leaflet of the lipid bilayer) (Figure 2), leading to Annexin-V binding, that is mainly associated with the microvesicles and, to a lower extent, with exosomes [82]. However, PS exposure also occurs in apoptotic bodies [83]; therefore, Annexin-V binding cannot be used to distinguish among different EV types.

Adding more complexity to exosome composition, it has been established that the same types of cells release distinct exosome subpopulations with unique compositions that may elicit differential effects on recipient cells [84]. Such exosome subpopulations possibly originate from different MVB subpopulations, and the dissimilar compositions of these subpopulations probably mirror the presence of multiple sorting machineries (i.e., ESCRT-dependent and independent, see above) that act on the MVB compartment [1] (Figure 1). Further investigation of exosome heterogeneity will improve our understanding of exosomal biology in health and disease. In this context, the most commonly used methods allow bulk analysis of vesicles (i.e., Western blot with EV markers), but are not competent for correct quantification and fail to uncover phenotypic heterogeneity in exosome populations. For such analyses, the development of multiparameter, high-throughput analysis methods at the single vesicle level is needed (i.e., [85,86,87,88]). Regarding lipid exosome composition, sphingomyelin concentration is higher in exosomes than in other vesicles, and LBPA is exclusive to endosomes and exosomes [81]. Inverted cone-shaped lipids like LBPA [61,89] in the external leaflet of the MVB lipid bilayer and ceramide, DAG, as well as PA in the internal leaflet [63], may facilitate negative curvature of the lipid bilayer needed for ILV formation and thus exosome formation [7,59,90].

Regarding the role of lipids in shedding vesicle formation, nSMase has been shown to be implicated in plasma membrane budding in several cell types [13,91]. nSMase inhibitors, or nSMase interference, reduce exosome secretion by blocking the ceramide-dependent inward ILV budding into MVB lumen [91]. In contrast, the nSMase blockade stimulates microvesicles/shedding vesicles budding at the plasma membrane [91]. Most probably, this is due to the asymmetric lipid distribution in the external and internal membrane leaflets, and thus sphingomyelin hydrolysis to ceramide might lead to more or less EV or exosomes depending on the membrane leaflet that is positively or negatively curved by sphingomyelin to ceramide conversion [90,92].

## 4. Extracellular Vesicles from T Lymphocytes

IS formation by T lymphocytes subsequent to TCR binding to antigen bound to MHC on the APC surface is a very dynamic, plastic and critical outcome involved in antigen-specific, cellular and humoral immune responses [93,94]. IS establishment integrates signals and combines molecular interactions leading to a proper and antigen-specific immune response [95]. There are two main groups of IS built by T lymphocytes leading to quite different, although essential, immune effector functions [93,95,96]. The interaction of helper T lymphocytes (Th), generally CD4^+^ cells, with MHC-II-bearing APC causes T lymphocyte activation (cytokine secretion, proliferation, etc.). In contrast, naïve CTL, usually CD8^+^ cells, recognize antigen-associated MHC-I on APC and become activated or “primed” to proliferate in the first phase, and kill target cells bearing the antigen in the second, effector phase. In the effector phase, primed CTL similarly form IS with target cells (virus-infected cells or tumor cells) leading to specific killing. Thus, the functional outcomes produced by the formation of an effective, mature IS include activation (naïve CTL and Th lymphocytes), killing (primed CTL), and functional anergy or apoptosis induction [97] once the effector phase is finished. IS formation induces the convergence of T lymphocyte secretion granules towards the MTOC and, almost simultaneously, MTOC polarization and secretion granules move towards the central supramolecular activation cluster (cSMAC) at the IS [94,98]. T lymphocyte secretion granules include cytokine-containing secretion granules in Th lymphocytes, cytolytic granules/secretory lysosomes in CTL, and MVB in Th lymphocytes and CTL. This dedicated mechanism appears to specifically endow the immune system with a superbly tuned tactic to enhance the efficiency of decisive secretory effector roles of T lymphocytes, while diminishing nonspecific, cytokine-controlled stimulation, target cell killing and activation induced cell death (AICD) of bystander cells [99].

CTL cytolytic granules are secretory lysosomes that have an MVB structure, and their degranulation causes ILV secretion as nanosize “extracellular vesicles” at the synaptic cleft made at the CTL-target cell interface, formerly described by Peters et al. [100]. Although CTL-secreted vesicles were not referred in this study as canonic exosomes, their creation and mode of exocytosis warrants this classification [10] (see above). Cytolytic granules contain perforin and granzymes that are located in a lumenal, electron dense core characterized by EM and also accumulate in the ILV [77]. In addition, it was shown that ILV and their derived exosomes contained, apart from the proapoptotic proteins perforin and granzymes, the exosome marker CD63 and molecules relevant for CTL-target cell interaction, such as TCR and CD8 [77,100], demonstrating that most of the cytotoxic factors exocytosed into the cleft between CTL and the target cell are membrane-enveloped or exosome-associated. However, release of perforin and granzymes in soluble form coming from the lumenal core cannot be excluded [77,101]. Perforin, which is inactive in the acidic environment of secretory lysosomes, is activated by neutral pH and Ca^2+^ at the synapse and polymerizes and forms a transmembrane pore that allows the entry of granzymes into the target cell; granzymes trigger caspase-dependent and independent cell death [44]. It was hypothesized that the presence of the TCR complex, CD8, and possibly other relevant molecules on these nanovesicles displaying their extracellular portions facing outwards, may ensure unidirectional delivery of lethal factors to target cells, since it was proposed that ILV released into the synaptic cleft bind specifically to the relevant antigen-MHC-I complex on the target cell membrane, and not to the CTL itself or to bystander cells [79]. This model would explain not only why a CTL does not kill itself, but also why bystander cells, which are in close proximity but do not bear the proper antigen, are not killed [79]. Subsequently, it was demonstrated that newly synthesized Fas ligand (FasL) is also stored in the limiting membrane of CTL secretory lysosomes and that polarized degranulation controls FasL delivery to the T cell surface, which is consistent with the role of a FasL-dependent pathway in CTL-mediated cytotoxicity [102].

In addition, it was shown that FasL can also be sorted from the MVB limiting membrane to ILV and hence to exosomes upon T lymphocyte activation, since T lymphocyte activation induced 100–200 nm “microvesicles” secretion including proapoptotic FasL and Apo2L [103] (Figure 2 and Figure 3). These microvesicles were subsequently characterized as canonic exosomes, since they arose from FasL^+^Apo2L^+^ ILV after MVB fusion with the plasma membrane [56]. Exosomal FasL and Apo2L, with the same topology as cell surface FasL and Apo2L, can bind to their respective death receptors on the surface of target cells, or effector T lymphocytes themselves, inducing caspase-dependent apoptosis [104,105] (Figure 3). Proapoptotic exosomes are thus involved in AICD of effector T lymphocytes, which constitutes an important suicide or fratricide mechanism participating in the downregulation of T cell-dependent immune responses [56,103,106]. Another major contribution was to demonstrate that inducible, polarized exosome secretion occurred at the IS formed by living Th lymphocytes and APC [18], as occurred in the IS formed by CTL [77,100] (Figure 3).

The hypothesis derived from all these publications that TCR activation of the effector T cell may induce the release of CD63^+^ exosomes bearing TCR was formally demonstrated by using TCR agonists to activate Jurkat Th, CTL and CD4^+^ lymphocytes [19,54]. Taken together, these reports constitute a major milestone in the exosome field since they demonstrate that TCR stimulation triggers inducible exosome secretion by T lymphocytes (both in CTL and Th cells) [2,12]. However, it is remarkable that, depending on the stimulation regime (absence or presence of co-stimulation signals), CD4^+^ T cell activation promotes the differential release of distinct EV subpopulations [110].

Adding more complexity to the EV field, CD63-enriched shedding vesicles or ectosomes directly budding from the Th lymphocyte plasma membrane and accumulating at the IS formed with a B lymphocytes acting as APC have been described [42] (Figure 3). These synapse-induced shedding vesicles were enriched in TCR and, upon endocytosis by APC, were capable of signaling via pMHC-II stimulation [42,111] (Figure 3), suggesting these synaptic ectosomes may facilitate the activation of B cells and other APC presenting the cognate pMHC-II. In this report, centrally accumulated TCR were located on the surface of extracellular microvesicles that bud at the IS centre. Members of the ESCRT-I family such as TSG101 sort TCR for inclusion in shedding vesicles, whereas VPS4 mediates microvesicle scission from the T-cell plasma membrane. TSG101 interference reduced EV production, whereas VPS4 function disruption rendered budding vesicles unable to undergo fission from the plasma membrane [42] (Figure 3). However, neither the existence of T cell-derived shedding vesicles in the CTL IS, nor a proapoptotic role for these EV, has been demonstrated yet. In addition, although nSMase has been shown to be involved in plasma membrane budding in several cell types [13,91], its participation in EV release from T lymphocytes has not been reported yet, in contrast with its role in ILV biogenesis [59].

## 5. Traffic of Cytotoxic Granules and MVB in T Lymphocytes

The molecular studies of CTL from patients with inherited defects leading to immunopathological haemophagocytic lymphohistiocytosis (HLH) syndrome, and their mouse models that impaired CTL function mediated by defects in secretory lysosomes/cytolytic granules, have underlined the significance of cytotoxicity in the control and termination of immune responses. The characterization of these alterations has improved our understanding of the key molecular events required for the maturation and traffic of cytolytic granules and the secretion of their cargo at the IS during target cell killing by CTL [44,112]. Included among the genetic disorders associated with the occurrence of HLH, mutations affecting lysosomal trafficking regulator (LYST) affect biogenesis of cytolytic granules [14], whereas AP3 mutations affect lytic granule polarization [25]. In addition, mutations in Rab27a affect docking of lytic granules to the synaptic membrane [26,113], and mutations in STX11 [41], MUNC13-4 [114] and MUNC18-2 [115] inhibit lytic granule degranulation by exocytosis. Perforin is critical for lytic granule-mediated cytotoxicity, as shown in reports from perforin-deficient mice and humans that carry hereditary mutation of perforin-encoding genes [116]. All of these genetic alterations produce alterations in CTL and NK cytotoxic function through the cytolytic granule-dependent pathway, resulting in severe HLH [44,112]. Not surprisingly, several regulators of cytolytic granules polarize traffic and fusion to the plasma membrane in T lymphocytes include some of the conserved molecular components of MVB traffic pathways, such as Rab27a [26] and Rab7 [27] (Figure 1). In addition, several members of the SNARE family, such as syntaxins 4, 7, 8 and 11 [38,41], have been described to be involved in lytic granule exocytosis in CTL, the latest stage of MVB traffic (Figure 1).

Regarding the molecular machineries specifically involved in MVB maturation and exosome biogenesis, and secretion in Th lymphocytes, there is little information available in comparison to other cell types, including CTL, probably due to the fact that, besides constitutive and non-polarized secretion, TCR triggering induces directional exosome secretion at the synapse [2,12,22], and this fact hinders experimental approaches. In fact, analysis of EV secretion at steady state, or upon TCR activation, may influence the mechanism of exosome secretion so that specific experimental conditions (steady state versus stimulation or costimulation) may affect the results [13]. Probably, the lipid pathway is the best characterized mechanism involved in exosome biogenesis and secretion in T lymphocytes. In this regard, silencing of nSMase2, or inhibition of its enzymatic activity, that decreases ceramide production induces a decrease in exosome release [15]. Some other lipid-metabolizing enzymes controlling T cell activation, such as diacylglycerol kinase α (DGKα) [117] that reduces DAG levels by producing PA, accumulate in the MVB limiting membrane and control MVB generation and CD63-enriched exosome release in T lymphocytes [17,18,19]. DAG, as well as ceramide and, to a lesser extent, PA, are cone-shaped lipids that when located in the inner leaflet of MVB limiting membrane can induce inward curvature and thus ILV formation [64]. Thus, pharmacologic inhibition of DGKα enhances MVB maturation and exosome secretion [18,19]. The finding that Hrs (ESCRT-0 family member) is not required for inducible exosome release by T lymphocytes [15] indicates that, although other ESCRT components have not been tested yet, exosome biogenesis seems to be ESCRT machinery-independent in T lymphocytes, unlike in other cell types [118,119]. Moreover, CTL from aSMase-deficient (aSMase-KO) mice are defective in exocytosis of cytolytic effector molecules upon IS formation. This defect results in attenuated cytotoxic activity of aSMase-KO CTL [21]. Thus, aSMase is required for contraction of secretory granules and expulsion of cytotoxic effectors, including exosomes, in CTL [21] (Figure 1). This reveals the importance of ceramide biosynthesis for both exosome biogenesis and the late stages of MVB traffic during inducible exosome secretion at the IS. Taken together, it appears that several components of the lipid pathways involved in ceramide and DAG/PA metabolism may be strong candidates for a pharmacologic or genetic intervention directed to modulate inducible exosome secretion by T lymphocytes.

Regarding the traffic of MVB and cytolytic granules towards the plasma membrane in T lymphocytes forming IS, MVB follow the microtubules network oriented by the MTOC, and exosome and lytic granule secretion is thus provided by the inducible polarized traffic of MVB and the MTOC towards the IS [31,120]. Therefore, it is remarkable that secretory polarized traffic leading to exosome secretion is an inducible and unique feature occurring in both CTL and Th lymphocyte IS upon challenge with antigen [12]. Thus, probably due to the intrinsic attributes of polarity and inducibility, pMHC-stimulated TCR evokes finely tuned, lipid-regulated specific signaling pathways leading to MVB polarization. In this context, there is a considerable number of studies on the characterization of the signals involved in MVB polarization, docking and fusion to the synaptic membrane, in comparison with the scarce reports on MVB maturation in T lymphocytes (reviewed in [12]). Exosome secretion is provided by the MTOC/secretory granules reorientation towards the IS in CTL and Th lymphocytes, which is initially guided by the DAG gradient concentrated at the IS, produced by TCR-stimulated phospholipase C (PLC) [16,94]. DAG phosphorylation by DGKα is implicated in the negative, spatiotemporal regulation of the DAG gradient [117] and MTOC/secretory granule orientation to the IS in CTL and Th lymphocytes [16]. DGKα inhibition enhances T cell activation by upregulating DAG levels [117]. DAG activates several members of the PKC, such as PKCθ and PKCδ, and protein kinase D (PKD) families [121]. PKCθ facilitates MTOC polarization by localizing dynein anchored to Adhesion and Degranulation Promoting Adapter Protein (ADAP) in the F-actin enriched areas at the IS, which pulls the MTOC forwards [35,36,122]. PKCδ is necessary for cytolytic granule polarization and cytotoxicity in mouse CTL [34] and for MVB polarization in Th IS [31]. In a Th-APC IS model, a positive function of TCR-triggered DAG, and a negative role of the DAG controller DGKα [117] in oriented MVB traffic towards the IS was shown [17]. These findings, together with the negative control that DGKα exerts on MTOC and cytolytic granule polarized traffic in CTL (described above) support that DAG, several DAG-activated PKC isoforms and DGKα can be contemplated as general regulators of secretory polarized traffic in T lymphocytes [12,20,35].

Much evidence supports an important role of actin cytoskeleton reorganization in exosome secretion triggered upon T lymphocyte IS (reviewed in [123]). Not surprisingly, several actin cytoskeleton regulators, such as actin-related proteins 2/3 (ARP2/3), hematopoietic lineage cell-specific protein 1 (HS1), transgelin-2 (TAGLN2), Diaphanous-related formin 1 (Dia1) and Formin-like 1 (FMNL1), have been shown to participate in MTOC-guided, polarized secretory traffic at the IS [124,125], although the contribution of all these actin cytoskeleton regulators specifically to exosome secretion has not been addressed yet. Since several recent reviews have dealt with the contribution of actin cytoskeleton reorganization to polarized secretory traffic at the IS [69,126,127], we focus on some relevant issues directly related to exosome secretion. F-actin depolymerization at the IS center controls MTOC and secretory granule polarization to the secretory domain of both CTL [30,120] and Th lymphocytes IS [128,129]. Regarding signals involved in actin reorganization, DAG-activated PKCδ is required for cortical actin reorganization at the IS, MTOC/MVB traffic to the IS and exosome secretion by Th lymphocytes IS [31]. Thus, cortical F-actin depolymerization at the central IS seems to be necessary to allow MVB docking to the plasma membrane for subsequent exosome release [13]. This supports that an altered actin reorganization at the IS may cause the defective MVB orientation occurring in PKCδ-interfered Th lymphocytes [31]. More recently, PKCδ-dependent depolymerization of the F-actin pool surrounding the MTOC/MVB (Figure 3) has been shown to be involved in MTOC/MVB polarization to the Th IS [32,33,69]. It is well known that a microtubule network is required for the transport of late endosomes, since MTOC co-migrates with diverse secretory granules (cytolytic granules/MVB in CTL, cytokine secretory granules/MVB in Th lymphocytes) [13,32,120]. Thus, crosstalk between actin remodeling and the microtubule network exists during polarized traffic at the IS [123], which would render the microtubule and/or actin cytoskeletons as pharmacologic targets to inhibit or to enhance exosome secretion. For instance, cortical F-actin cytoskeleton depolymerization may indirectly promote exosome secretion. However, due to the involvement of the actin and tubulin cytoskeleton in a myriad of crucial cellular functions, targeting these networks is also probable to provoke miscellaneous non-specific effects [13].

In summary, pharmacologic or genetic DGK inhibition might be used therapeutically to enhance inducible exosome secretion by T lymphocytes [12]. DGKα inhibition also enhances DAG levels at the early stages of T cell activation [117]. Thus DGKα inhibition, by increasing both T cell activation and inducible exosome secretion [17,18,19], boosts exosome secretion and might be useful for cell-free, exosome-based therapies (see below).

## 6. Chimeric Antigen Receptor (CAR) T Cells and CAR T Cell-Derived EV

### Cancer Therapeutic Approaches

The role of EV from immune cells, including T lymphocytes, in anti-tumor immunity has been recently and exhaustively reviewed [130,131]; therefore, we focus on potential therapeutic uses of CAR T cell-derived EV. CAR T lymphocyte-based immunotherapy has proven to be a promising treatment of patients suffering several refractory cancer diseases [132]. CAR consists of an extracellular domain that confers antigen-recognition specificity, a transmembrane domain, and an intracellular signaling domain (CD3ζ) that provides activation signals to T lymphocytes [132]. However, several challenges preclude the use of adoptively-transferred CAR T cells, including their low efficacy against solid tumors, immunosuppression by tumor microenvironment, poor T cell persistence, T cell dysfunction or exhaustion, cytokine release syndrome (CRS) [132,133] and immune effector cell-associated neurotoxicity syndrome (ICANS). CRS is associated with supraphysiologic cytokine production and massive in vivo T cell expansion [134], whereas the cause of ICANS remains poorly understood, although appears to be related with direct central nervous system toxicity by the CR T cells, diffusion of inflammatory cytokines through the blood-brain barrier and the disfunction of this barrier caused by CAR T cells and/or cytokines [135]. It is out of the scope of this review to summarize the preclinical and clinical uses of CAR T cells for cancer therapy (there are more than 250 clinical trials testing CAR T cells; please refer to recent and superb reviews on this topic [132,133]), thus we will focus on trials using CAR T cell-derived EV.

Considering the early hypothesis that the presence of both TCR and proapoptotic molecules (FasL, Apo2L, perforin, granzymes A and B) on T cell-derived exosomes would confer on them both antigenic specificity and guiding cytotoxicity [77,79,100], making them potent vectors to deliver proapoptotic cues to target cells bearing the cognate antigen, several strategies have been developed to test the use of CAR T cell-derived exosomes for cancer therapy. The fact that TCR activation boosts the secretion of CTL-produced exosomes, and the presence of the TCR/CD3ζ complex in these exosomes (see above) [54] would reinforce this approach. TCR/CD3ζ complexes endocytosed after recognition of the pMHC-II complexes are targeted to MVB, then to ILV and hence exosomes [54]. A crucial prerequisite for using CAR T cell-derived exosomes to specifically induce tumor cell death is the presence on exosomes of the CAR molecule, since its antibody-derived, antigen-binding variable fragment endows them with tumor cell specificity, as occurs with CAR T cells [132,136]. Since CAR is an artificial molecule containing the CD3ζ intracellular signaling and localization domain, it was unknown whether CAR was present in exosomes and shedding vesicles, and whether CAR conferred a specific cytotoxic effect, until recent reports characterized CAR expression and function in exosomes [55] and shedding vesicles [137]. This prerequisite has been endorsed in several preclinical studies by using exosomes and/or EV produced by CAR T cells [55,80,138] (Table 1).

CAR T cell-derived exosomes exhibit excellent capability for use as direct aggressors in immunotherapy, since ex vivo-produced human exosomes transporting human EGFR and HER2-specific CAR have powerful in vivo activity against EGFR^+^ and HER2^+^ human tumor cells in xenograft models [55]. These CAR T cell-derived exosomes specifically induce apoptosis of tumor cells expressing the antigens recognized by CAR on the cell surface but do not kill tumor cells that do not express these antigens [55]. Comparable results have been obtained using HEK-derived exosomes, showing that only when exosome entry into cells is mediated via binding to the CD19 antigen on the surface of CD19^+^ B-cells does pro-apoptotic signaling occur resulting in selective cytotoxicity [80,138]. Certainly, the evoked apoptotic mechanism does not depend on FasL, Apo2L, perforin and granzymes, as for CAR T-cell derived exosomes, since HEK293 cells do not express these molecules. More recently, an interesting paper demonstrated an in vivo role of the RNA component of signal recognition particle 7SL1 (RN7SL1, a non-coding RNA that activates interferon-IFN-stimulated genes) contained in EV from CAR T cells in tumor rejection [140]. In this approach, the CAR construct contains downstream of the CAR sequence a U6 promoter that drives the transcription of human RN7SL1 and, optionally, a 5′LTR promoter driving the transcription of a peptide antigen. The authors demonstrate that RN7LS1-containing CAR T cells deliver RN7SL1 in EV in vivo, that orchestrates endogenous immune activation to improve responses against the tumor. EV released by CAR T cells upon CAR engagement transfer RN7SL1 to endogenous immune cells (myeloid cells, DC and T cells), but not to the tumor cells, and EV release and RN7LS1 transfer is inhibited by a nSMase inhibitor. RN7LS1 inside target cells activates a RIG-I-dependent, IFN-mediated inflammatory response and induces transcription of IFN-stimulated genes. RN7LS1 delivery in EV to immune cells improves immunostimulatory properties of myeloid and DC cells that, in turn, effectively activate the function of endogenous CD8 T cells against the tumor. All these immune cells, acting together, may trigger solid tumor rejection even in case of CAR-recognized antigen loss by the tumor. Although in this research neither EV nature or composition, nor the presence and contribution of CAR and proapoptotic proteins in the EV to tumor rejection, have been established, this strategy opens new venues based on improved endogenous immunity against tumor cells conferred by EV, since CAR T cells can now co-deploy antigenic peptides with RN7SL1 released by EV to enhance their efficacy against tumor cells, even when tumors lack adequate neoantigens [140]. Table 1 summarizes the most relevant features of the preclinical trials involving CAR T cell and HEK293-derived EV. Most of these trials have been performed ex vivo or in vivo using xenograft models in mice.

The advantages of CAR T cell-derived exosomes as cell-free immunotherapy are their independence of CAR T cell life span and division, their stability, some obvious logistic issues, the low risk of collateral toxicity (i.e., CRS incidence) when contrasted to CAR T cells, and the fact that exosomes lacking PD-1 (in contrast to PD-1-expressing T cells) are refractory to PD-L1 immunosupression by the tumor [55] (Table 2). In this context, it is remarkable that CAR T cell-induced CRS is one of the most harmful complications that follow infusion of the CAR T cells and occurs in approximately two thirds of CAR T cell recipients, generally within 10 days after cell infusion. This life-threatening complication is generally ascribed to uncontrolled release of cytokines from CAR T cells [136]. Moreover, exosomes could be distributed through the blood circulation and other body fluids as supported by the abundance of exosomes found in most body fluids. In addition, exosomes have the ability to cross certain biological barriers, such as the blood–brain barrier and blood-tumor barrier, as documented by the presence of tumor cell-originated exosomes in body fluids [136] (Table 2).

The fact that such a cell-free “exosome therapy” involves the on-bench stimulation and expansion of the effector CAR T cells [55] may allow the genetic modification and/or pharmacologic treatment of the cultured cells to increase T cell activation, and/or exosome biogenesis, to enhance EV production. In addition, exosome collection and purification ex vivo involve the elimination of effector CAR T cells, and also bystander or contaminating cells, before exosome infusion [55]. Thus, this strategy would circumvent the undesirable possibility of transducing, for instance, CAR to residual tumoral cells during T cell manufacturing that may lead to provocation of resistance to CAR T cell therapy by unintentional transduction of a single leukemic B cell, as reported in [141]. These findings illustrate the need for purging residual contaminating tumor cells from engineered CAR T cells or using alternatives such as cell-free, exosome-based therapies.

Taking into account that it is not known whether CAR signaling or the involvement of any of the exosomal proapoptotic molecules (perforin, granzymes, FasL, etc.) mediate the therapeutic effects described, these findings require further confirmation, and to formally establish the contribution of these molecules, although they convincingly support the use of exosomes as biomimetic nanovesicles in antitumor therapy [55]. Recent reviews have dealt with the proapoptotic mechanisms evoked by the proapoptotic molecules present in EV on target cells; please refer to these for further details [130,142].

It is remarkable that most of the strategies directed to unveil the role of the molecular components involved in exosome biogenesis/degradation and/or release in T cells (ESCRT, tetraspanins, Rabs, MAL, ISGylation, SNAREs, etc.) are mainly based on either the expression of dominant negative mutants or RNA interference, which leads, in a vast majority of the approaches, to EV secretion inhibition [6,64]. Thus, although these approaches have been useful in establishing the necessity of diverse components for exosome secretion, very few of these approaches have led to an increase of exosome secretion [17,18,19,64,143]. In principle, the positive modulation of exosome secretion ex vivo could be a useful strategy to enhance the effectiveness of exosomes for subsequent in vivo applications. Among all the potential approaches designed to enhance on bench exosome secretion, perhaps the regulation of the lipid pathways and their metabolites (DAG, ceramide) involved in exosome biogenesis appears the most feasible approach, due the existence of quite specific pharmacologic agents suitable for exosome induction ex vivo. Thus, increasing DAG levels using DGK inhibitors (i.e., R59949) [19], or ceramide levels using sphingomyelin synthase inhibitors (i.e., D609) [60], which have been shown to increase exosome secretion, may constitute useful tools for this strategy.

Although the timing and modes of activation and maturation are different, both CTL and Natural Killer cells (NK) utilize an overlapping arsenal consisting of cytotoxic effector proteins including FasL [102], perforins, granzymes and granulysin contained in their secretory lysosomes [78,144] (recently reviewed in [142]). The molecular mechanisms controlling the maturation and traffic of the secretory lysosomes and the polarized secretion of exosomes towards the synapse are, in great part, common to both cell types [44,102,145]. Thus, it is conceivable that approaches directed to increase exosome secretion in CTL ex vivo can also be useful to boost exosome secretion by CAR NK cells. However, although TCR activation enhances the constitutively low secretion of exosomes by T lymphocytes, resting NK cells secrete pro-apoptotic exosomes with no differences in the amounts of exosomes or marker expression relative to activated NK cells via NK activating and inhibition receptors [78,146]. NK cell expansion ex vivo increases exosome secretion [146], as occurs in T lymphocytes, which constitutes an useful strategy to produce on a large scale exosomes that may lead to new preclinical and clinical applications [146]. Different to T lymphocytes, NK cell recognition of their targets is not controlled by antigen specificity but rather through the integration of signals evoked by activating and inhibitory receptors, activated by a myriad of ligands in the target cells, which requires a deeper knowledge of the complex NK biology and signaling before any experimental design. This fact probably has led to CAR NK therapy being less developed than CAR T cell therapy in general [147], and therefore there are less pre-clinical trials using CAR NK-derived EV in particular [146]. However, the use of CAR NK cells has some advantages (and disadvantages) when compared with CAR T cells [147]. Although the use of CAR NK cells in clinical trials has been less extended in comparison to CAR T cells, in principle it is conceivable the use of NK-derived EV in future anti-tumor therapies.

## 7. Future Developments in the Field and Concluding Remarks

The application of CAR T cell-derived EV will make this modality of cell-free-based therapy more clinically controllable, avoiding CRS syndrome and allowing the tuning of EV production by the CAR T cells. However, EV treatment should be considered as a complementary rather than a substituting technique for CAR T cell-based therapy [80]. Remarkably, recent in vivo results support that RN7SL1-containing CAR T cell-derived EV may be transferred to endogenous immune cells and synergize with CAR T cells to enhance their efficacy against tumor cells, even when tumor cells lose CAR antigen expression, by activating the endogenous immune system [140]. In addition, it will be necessary to unveil certain unclear aspects regarding EV bioactive cargo and several mechanistical issues. One relevant issue is to address the exact nature (shedding vesicles and/or exosomes) of the EV involved in the observed apoptosis of tumor cells. As summarized in Table 1, in most of the cases this issue has not been addressed. Since the pathways for shedding vesicles and exosome secretion share several regulatory components (i.e., ESCRT, nSMase, ceramide), and some of them act in the opposite direction (nSMase and ceramide have positive roles in exosome secretion in contrast to their negative role in shedding vesicle secretion), it is necessary to clarify the exact nature of the EV responsible for the observed effects in order to optimize ex vivo induction and production of the bioactive EV, and to enrich in the relevant EV type or to eliminate bystander EV. Second, it is necessary to establish the nature of the diverse proapoptotic cargo molecules in the EV responsible for apoptosis [130,142] and the evoked apoptotic mechanisms in the target tumor cells, or their immunostimulatory role [140,148], since in the published reports to date using CAR T cell-derived exosomes all these aspects have not been formally clarified (Table 1). With this knowledge in hand, it may be possible to design strategies to modify and/or enhance the EV bioactive cargo [140,148].

Third, the fact that the same types of cell release distinct exosome subpopulations [110] with distinct compositions may cause differential effects on recipient cells [84]. Further characterization of exosome heterogeneity, together with the identification of the proapoptotic and bioactive cargo molecules, will enhance our knowledge of exosomal biology in health and disease. To this end, the development of multiparameter, high-throughput, single-vesicle-based analytical and preparative methods is needed [85]. Indeed, with this knowledge in hand it will be possible, for instance, to enrich the exosome preparations in the exosome subpopulation carrying the bioactive, relevant molecule. NTA, one of the most commonly used technique for EV analysis, does not allow at this moment the high throughput and the multiparametric, simultaneous acquisition of several fluorochrome emissions. Currently, NTA instruments with two different fluorescent channels have been developed to study two fluorescent probes in the same EV preparation (but not simultaneously in the same vesicle) [87], but lack multiparametric and high throughput capabilities. In principle, flowcytometry (FC) is an ideal technique for EV analysis due to its intrinsic high throughput, multiparametric characterization capability, robust statistical analysis of individualized EV and EV preparative potential, but most conventional FC platforms suffer from detection thresholds above 500 nm [149]. To circumvent this, a dedicated FC has been developed using optimized, laboratory-built configurations of a commercially available high-end FC [85]. Furthermore, using an optimally configured and customized commercial flow cytometer, fluorescence-activated vesicle sorting (FAVS) was established as a method to analyse and sort exosomes based exclusively on the presence of endogenous membrane components, including EGFR and the exosomal-enriched marker, CD9 [150]. FAVS represents a major technical improvement by allowing analysis and sorting of exosomes founded on the expression of selected cell-surface markers. The analysis and purification of EV via FAVS will expand our knowledge of these EV as well as their subpopulations, which are altered in diseases, and may help to unveil the molecular mechanisms causing their biological function [150]. However, complicated manual hardware adaptations, adjustments and calibration prior to use are needed [85,151]. This fact precludes the generalized use of dedicated FC to EV sorting and analyses. Thus, alternative, recent implementations have been developed including image FC (iFC) and nano FC. Nano FC allows single vesicle detection down to 7 nm size, and the incorporation of multiparameter fluorescence detection allows sizing and profiling of individual EV as small as 40 nm [152]. Certain advanced image techniques such as single-molecule localization microscopy (SMLM) [99] may complement nano FC due to SMLM superb EV detection sensitivity and resolution, although this technique fails to determine EV size and concentration [153]. In addition, high resolution image FC (iFC) [154] facilitates the robust detection and quantification of phenotypically distinct, single EV in heterogeneous samples by using different fluorochrome conjugated antibodies [155], and constitutes a new and encouraging approach for EV analyses. Please refer to excellent reviews summarizing and comparing the different EV analysis methods [153,156].

Fourth, it is remarkable that engineered CAR T cells are cultured during 7–9 days in the presence of interleukin-2 to expand the culture, after the initial stimulation with anti-CD3/CD28 to facilitate CAR transduction [55,80,138]. Subsequently, the induction of exosome secretion requires restimulation of the expanded CAR T cells with antigen-expressing cells or anti-CD3/CD28. The schedule of the second activation is based on the necessity to return to the resting activation state of CAR T cells, to minimize AICD [55]. In this context, FasL-carrying, exosome-dependent AICD occurs when activated T lymphocytes undergo a too early restimulation [19,56], which could be a problem since the number of exosome-producing CAR T cells would decrease and thus exosome production. Considering these facts, an adequate timing for restimulation, and the enhancement of T cell activation at this stage, together with positive modulation of exosome biogenesis will potentially be a synergizing strategy for ex vivo exosome production. For instance, the enhancement of CAR T cell activation achieved by DGK inhibition has been proposed as a promising approach to enhance the antitumor efficacy of CAR T cells due to the enhanced anti-tumor effector functions [133]. Moreover, the fact that cell-free “exosome therapy” involves on-bench stimulation and growth expansion of the effector CAR T cells [55] may allow the genetic modification and/or pharmacologic treatment of the effector cells in culture, directed to increase T cell activation and/or to enhance EV production. In this context, we propose to transduce genes and/or to add pharmacologic enhancers of T cell activation during the restimulation as a novel strategy to enhance EV production. Since some T lymphocyte activation enhancers are also positive modulators of exosome biogenesis and secretion in T lymphocytes (i.e., DGK inhibitors such as R59949) [17,18,19,117], they may constitute a promising strategy for cell-free, CAR T cell-derived exosome-based therapies.

Fifth, it is becoming clear that the presence of endogenous or adoptively transferred T lymphocytes harboring a stem-like memory or a precursor memory phenotype correlates with improved therapeutic results in adoptive cell therapy [133]. In order to guarantee this principle, one promising approach consists in using precursor T cells or antigen-experienced memory T cells for CAR transduction [133,157]. Unfortunately, any EV-based, cell-free strategy will not directly confer immunological memory unless the ex vivo-produced exosomes are inherently capable to induce T cells immunostimulation in the recipient. While dendritic cells and B lymphocyte-produced exosomes contain MHC-I and MHC-II, and can efficiently present antigen and subsequently stimulate a T-cell memory response [3,47,158], T cell-derived exosomes in general, and CAR T cell-derived EV in particular, have not been characterized yet as competent immunostimulatory, memory-inducer entities [159] (Table 2). To circumvent this problem, the repetition of CAR T cell EV infusion to the recipient, the co-delivery with dendritic cells-derived EV previously pulsed with the targeted antigen and/or to arm the CAR T cell EV with co-transduced, immunostimulatory molecules, including antigenic peptides (see below), may constitute useful strategies for the future.

In addition, although CAR T cell-based adoptive cell therapy has achieved success in the treatment of B cell malignances [132], this approach has demonstrated variable or low therapeutic efficacy in solid tumors [133]. This could be due to several factors including low persistence and expansion of CAR T cells in the recipient, tumor antigen escape (both are factors concurring in B cell malignances), together the insufficient ability of CAR T cells to traffic and infiltrate solid tumors, as well as the immunosuppressive tumor microenvironment (i.e., expression by the tumor of immune checkpoint PD-L1) and physical tumor barriers [132,135]. It is expected that all these life-threatening factors, together with the above mentioned toxicities intrinsic to CAR T cell therapy (CRS, ICANS) [135], will be greatly ameliorated by using CAR T cell derived EV (Table 2).

Finally, recent results showing that immunostimulatory RN7SL1 and antigen peptide released in EV by CAR T cells can be operated to enhance an endogenous immune response against tumors opens the possibility that CAR T cells can be used to deliver EV armed with RN7SL1 or RN7SL1 together a peptide antigen of choice [140], provided that the neoantigen is present in the tumor. Thus, EV from CAR T cells may transport immunostimulatory RN7SL1 or other immunorelevant molecules, such as immunogenic peptides than can stimulate specific memory T cells to kill tumors, to the tumor microenvironment. A better understanding of the biology of EV secretion, composition and targeting and uptake by target cells is needed (Table 1), exemplified by the unexplained observation of biased RNA delivery to endogenous immune cells but unbiased delivery of peptide antigens to both tumors and immune cells in the tumor microenvironment [140,148]. In addition, CAR T cells could be armed during CAR transduction with additional proapoptotic or other bioactive molecules that could be delivered by EV. Most human cancers lack adequate neoantigens, potentially limiting the benefit of CAR T cells that can recruit an endogenous immune response. Thus CAR T cells delivering immunostimulators could be armed with selected peptide antigens, or proapoptotic molecules to potentially overcome major barriers impeding CAR T cell efficacy against solid tumors [140].

## Figures and Tables

**Figure 1 cells-11-00790-f001:**
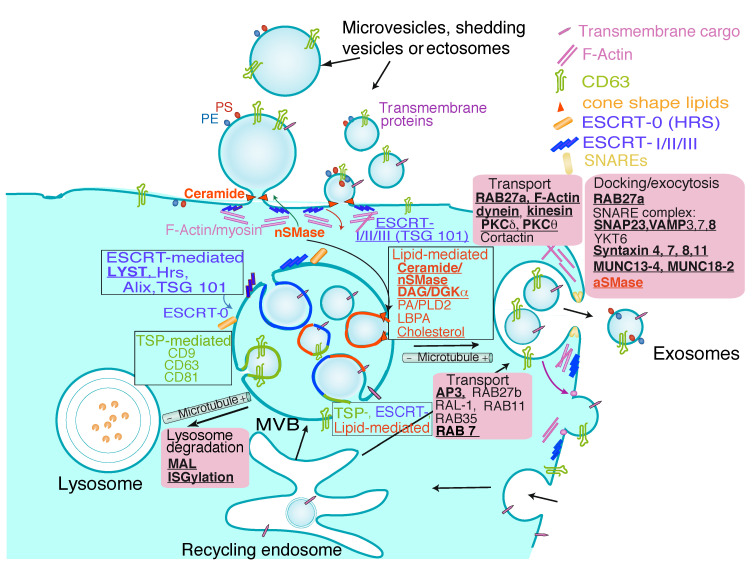
Extracellular vesicles. EV of different intracellular origins can be secreted by eukaryotic cells. The figure represents the different types of vesicles released, either by direct budding from the plasma membrane or by generation of ILV inside MVB, that subsequently fuse with the plasma membrane releasing exosomes. Apoptotic bodies released by dying cells have been excluded. For clarity’s sake, only the constitutive secretion of EV and exosomes is represented, although in certain immune cells such as T and B lymphocytes the traffic of MVB and the secretion of exosomes can be induced via T cell receptor (TCR) and B cell receptor (BCR) stimulation [12]. Traffic of MVB comprises three general phases: ILV biogenesis during the maturation of MVB, transport of MVB to the plasma membrane and docking and fusion of MVB to the plasma membrane, whereas EV secretion involves a single step. Transport and fusion of MVB to the lysosomes may lead to MVB degradation. For more details, please refer to [1,6,13]. General and T lymphocyte-specific mechanisms of shedding vesicles and exosome biogenesis and MVB traffic are represented. The inward, intraluminal budding of specific membrane nanodomains from the MVB limiting membrane produces ILV. The invagination of ILV and the sorting of specific cargoes can be produced by the action of three mechanisms that are enclosed in black line boxes: Endosomal Sorting Complex Required for Transport (ESCRT)-0-I-II-III machinery (blue), tetraspanins (TSP) (green) or certain lipids as cholesterol, ceramide, diacylglycerol (DAG) and lysobisphosphatidic acid (LBPA) (red). In addition, multiple machineries (represented as mixed colors) can collaborate in ILV biogenesis. It is unclear whether the three mechanisms act simultaneously on the same MVB or each one acts on different MVB, although all mechanisms are shown operating in the same MVB for clarity’s sake. Black line rectangles enclose the general mechanisms involved in exosome biogenesis, whereas the regulators of MVB traffic (including transport to lysosomes for degradation, transport to the plasma membrane, docking and fusion with the membrane) are enclosed in magenta boxes. ESCRT-0 components (hepatocyte growth factor-regulated tyrosine kinase substrate -Hrs-, STAM) are generally not observed in plasma membrane budding leading to shedding vesicles, whereas ESCRT-I-II-III are involved in these processes (reviewed in [13]). However, both ESCRT-0 and ESCRT-I-II-III are involved in ILV formation inside MVB [1,13]. Actin cytoskeleton depolymerization is required for secretion of shedding vesicles and exosomes. In addition, externalization of phosphatidylethanolamine (PE) and phosphatidylserine (PS), that binds Annexin V, occurs in plasma membrane-derived EV and, to a lower extent, in exosomes. Bold, underlined characters identify those molecular components or processes that regulate MVB secretory traffic in T lymphocytes: lysosomal trafficking regulator (LYST) [14], neutral sphingomyelinase 2 (nSMase2) [15], DAG [16], diacylglycerol kinase α (DGKα) [17,18,19,20], acidic sphingomyelinase (aSMase) [21], MAL [22,23], ISGylation [24], Adaptor protein 3 (AP3) [25], Rab27a [26], Rab11, Rab7 [27], dynein [28], kinesin-1 [29], cortical F-actin [30,31], centrosomal area F-actin [32,33], protein kinase C δ (PKCδ) [31,34], protein kinase C θ (PKCθ) [35,36], vesicle-associated membrane protein 8 (VAMP-8) [37], syntaxin 4 (STX4) [38], syntaxin 7 (STX7) [39], syntaxin 8 (STX8) [40], syntaxin 11 (STX11) [41], SNAP23 [38]. Underlined characters identify molecules involved in shedding vesicles generation in T lymphocytes: tumor susceptibility gene 101 (TSG101) and vacuolar protein sorting 4 (VPS4) [42]. LYST has Hrs (an ESCRT-associated protein) as binding partner, which supports that LYST participates in MVB biogenesis [43,44].

**Figure 2 cells-11-00790-f002:**
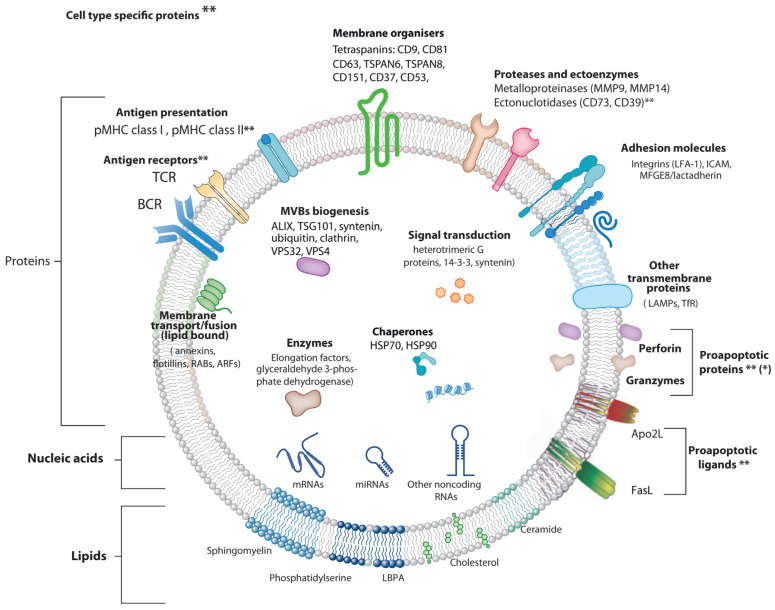
Exosome structure and molecular composition. Exosomes are surrounded by a phospholipid bilayer and contain nucleic acids and proteins (grouped by biological function), lipids, and nucleic acids. Exosomal proteins include annexins, important for transport; tetraspanins and integrins important for cell targeting and binding, and Alix and TSG101, involved in exosomal biogenesis from endosomes. Abbreviations: FLOT1, flotillin1; HSP, heat shock protein; MHC, major histocompatibility complex; RabGDI, RabGDP-dissociation inhibitor; RAP1B, Ras-related protein1B; TSG101, tumor susceptibility gene 101. (**) labels those proteins that are specifically found in exosomes produced by T or B lymphocytes, whereas the rest of the indicated proteins are mostly found with high frequency (>30%) in exosomes produced by different cell types [2]. (*) indicates it is not clear whether perforin/granzymes are located or not inside exosomes. For more details regarding exosome composition visit http://www.exocarta.org, accessed on 24 January 2022.

**Figure 3 cells-11-00790-f003:**
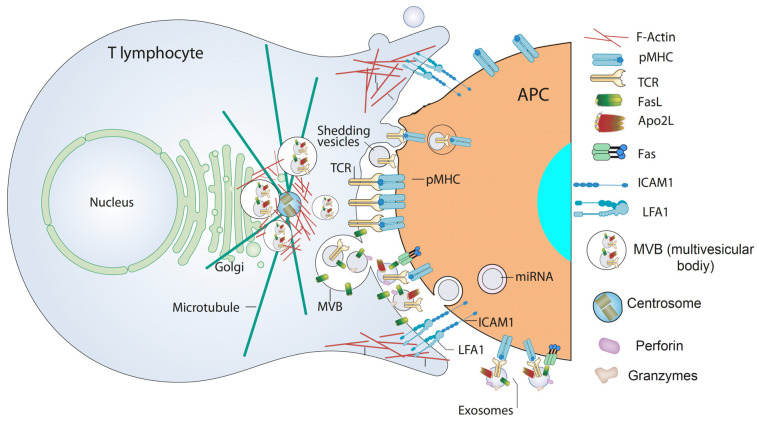
EV in the immune synapse. In a mature IS produced by TCR stimulation via the peptide-MHC complex (pMHC) on the APC and the interaction of accessory molecules (such as Intercellular Adhesion Molecule 1—ICAM1—with Lymphocyte function-associated antigen 1—LFA-1) F-actin is reduced at the cSMAC, the central region of the IS. F-actin accumulates at the distal SMAC (dSMAC), and F-actin around the centrosome depolymerizes. These F-actin reorganization processes, acting in a coordinated manner, may assist centrosome traffic towards the IS and the simultaneous convergence of MVB towards the F-actin depleted area in the cSMAC, facilitating MVB fusion at the cSMAC, and the subsequent exosome secretion carrying TCR and proapoptotic molecules in the synaptic cleft. In addition, shedding vesicles emerging from the plasma membrane and containing TCR are represented at the synaptic cleft. Both exosomes containing miRNA [15] and shedding vesicles [107,108] are engulfed by APC and provide biological responses in APC. For more details please refer to [12,69,107,109].

**Table 1 cells-11-00790-t001:** CAR T cell and cell-derived EV: preclinical studies.

Target Molecule	EV-Producing Cell	EV Types	EV Phenotype	Anti-Tumor Mechanism	Target Cell
EGFR, HER2 [55]	Human CAR T cells (?) ^1^	Exosomes	CAR^+^, CD3^+^, CD63^+^, perforin^+^, granzyme B^+^, CD45^−^, CD28^−^	Perforin/ granzyme B ^2^	EGFR^+^, HER2^+^ human breast cancer cells
HER2 [138]	Human CAR T cells CD4^+^ (46%) CD8^+^ (49%)	EV (small EV, probably exosomes plus larger EV) ^3^	CAR^+^, CD3^+^, CD63^+,^ granzyme B^+^	Granzyme B ^2^	HER2^+^ human breast cancer cells, ovarian cancer cells
Mesothelin [80]	Human CAR T cells CD4^+^ (58%) CD8^+^ (31%)	Probably exosomes ^4^	CAR^+^, CD3^+^, CD63^+^, perforin^+^, granzyme B^+^	Perforin/ granzyme B ^2^	Triple negative human breast cancer cells
CD19 [139]	Human CAR HEK293 cells	Probably exosomes ^4^	CAR^+^, CD63^+^, CD81^+^	Indirect induction of proapoptotic genes in target cells	CD19^+^ human B cell leukemia
CD19 [137]	Human CAR HEK293 cells	Probably shedding vesicles ^4^	CAR^+^, annexin V binding (PS exposure)	MYC Gene disruption mediated by CRISPR/Cas9	CD19^+^ human B cell leukemia cell lines
Mesothelin CD19 [140]	Human and mouse CAR T Cells (?) ^1^	EV ^4^	Unknown ^5^ Contain RN7SL1	Recruitment of endogenous anti-tumor immunity byRN7SL1	Mouse melanoma expressing human CD19

^1^ No data available regarding CD4^+^/CD8^+^ subpopulations. ^2^ Proposed mechanism, but not formally demonstrated. ^3^ Unfractionated EV; more criteria to differentiate exosomes from shedding vesicles are needed. ^4^ More criteria to differentiate exosomes from shedding vesicles are needed. ^5^ No phenotypic analyses of EV were performed.

**Table 2 cells-11-00790-t002:** CAR T cell and CAR T cell-derived EV. A comparison.

Event	CAR T Cells	CAR T Cell-Derived EV
Cytokine releasing syndrome	++	−
Neurotoxicity	++	−
Cross the blood barrier	−	++
Efficiency against solid tumors	+/−	++
Immunosuppression by tumoral PD-L1	+	−
Immunological memory	+ ^1^	(?) ^2^

^1^ Depends on the use of central memory or effector memory CAR T cells. ^2^ Not formally established.

## Data Availability

Not applicable.

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
