# Peer review of "T Lymphocyte and CAR-T Cell-Derived Extracellular Vesicles and Their Applications in Cancer Therapy"

_cells, 2022, doi:10.3390/cells11050790_

Round 1
Reviewer 1 Report
Useful and timely review, of interest to both clinicians and basic researchers. However, its structure and content can be improved.
The introduction section (8 printed pages) dealing with exosome preparation, characterization, biogenesis, signaling, etc does not reflect the title of the paper and it is somehow confusing. With respect to the mechanisms, there is also some redundancy with section 4. Please address these concerns.
Notably, the induction of T cell memory is critical for the success of immunotherapy using various therapeutic approaches. CAR- T cells with a central memory and stem cell memory can be generated. So, what is the impact of T lymphocyte-derived exosomes on the induction of T cell memory? Please comment on this.
In Page 18, the authors mentioned that the delivery of CAR-T cells may be armed with neoantigens. But, if the tumors lack the used neoantigens, I don't see how this can enhance the benefit of the mentioned therapy.
The authors should consider the addition of a short paragraph on exosome released from CAR-NK cells. Also, add a table describing the advantages and disadvantages of using exosomes as compared CAR-T cells, which suppose to release exosomes upon in vivo activation.
Author Response
Reviewer 1
Useful and timely review, of interest to both clinicians and basic researchers. However, its structure and content can be improved.
The introduction section (8 printed pages) dealing with exosome preparation, characterization, biogenesis, signaling, etc does not reflect the title of the paper and it is somehow confusing. With respect to the mechanisms, there is also some redundancy with section 4. Please address these concerns.
Referee is right and Referee No. 3 raised a similar caveat. Accordingly, we have changed the title. We include “T lymphocyte-derived extracellular vesicles” (and not only exosomes) since we deal with the molecular bases involved in EV generation in general (exosomes and shedding vesicles), in T lymphocytes. In addition, since the exact nature of the EV (exosomes and/or shedding vesicles) involved in CAR T cells effects is somewhat ambiguous (see Table 1) we have used the inclusive term “extracellular vesicles” in the title, for coherence.
Regarding the amplitude of the introduction, we feel that the knowledge of the molecular bases involved in EV (exosomes and shedding vesicles) generation and secretion T lymphocytes, and the description of EV molecular composition (presence of proapoptotic molecules, TCR and CAR, etc.), are crucial to understand how CAR T cells produce antigen-specific, bioactive EV, and how this can be tuned to improve EV-based therapy. That is the reason we dedicate several pages to this important introductory issue. In addition, since the exact nature of the EV (exosomes and/or shedding vesicles) involved in CAR T cells function remains unclear (table 1), we develop a thorough description of the EV types, normalization attempts and isolation protocols. However, to satisfy this referee’s point, we have balanced the introductory sections with the rest of the manuscript, and deleted the somewhat redundant figure 1 (also suggested by the referee No. 2) and some spare redundant sentences, and we have included more content in Section 6 and 7, as requested also by referee No 3.
Notably, the induction of T cell memory is critical for the success of immunotherapy using various therapeutic approaches. CAR- T cells with a central memory and stem cell memory can be generated. So, what is the impact of T lymphocyte-derived exosomes on the induction of T cell memory? Please comment on this.
Referee is right, and we have introduced a new paragraph in Section 7 to deal with this important point.
In Page 18, the authors mentioned that the delivery of CAR-T cells may be armed with neoantigens. But, if the tumors lack the used neoantigens, I don't see how this can enhance the benefit of the mentioned therapy.
Referee is right, and we have included a new statement to clarify this important issue.
The authors should consider the addition of a short paragraph on exosome released from CAR-NK cells. Also, add a table describing the advantages and disadvantages of using exosomes as compared CAR-T cells, which suppose to release exosomes upon in vivo activation.
Referee is right; we have included some sentences and a new table in Section 6 to satisfy referee’s points.
Reviewer 2 Report
The paper of Calvo and Izquierdo summarized the T lymphocyte-derived exosomes and their applications in cancer therapy. This review is a comprehensive review. They gave a background on extracellular vesicles types, normalization attempts, and isolation protocols. Then, exosome biogenesis, composition, and regulation of exosome secretion. Further, they mainly reviewed the EVs from T lymphocytes, their anticancer cargoes, and CAR T cell-derived EV for cancer therapy. The review is well planned and written manuscript.
Regardless of the scientific significance of the publication. I have raised some comments:
Comment 1: Figure 1 does not bring anything new to work, authors may consider removing it.
Comment 2: Figure 2 letters are too small, please increase its font size.
Comment 3: Editing of the English language and style are required.
Author Response
Reviewer 2
The paper of Calvo and Izquierdo summarized the T lymphocyte-derived exosomes and their applications in cancer therapy. This review is a comprehensive review. They gave a background on extracellular vesicles types, normalization attempts, and isolation protocols. Then, exosome biogenesis, composition, and regulation of exosome secretion. Further, they mainly reviewed the EVs from T lymphocytes, their anticancer cargoes, and CAR T cell-derived EV for cancer therapy. The review is well planned and written manuscript.
Regardless of the scientific significance of the publication. I have raised some comments:
Comment 1: Figure 1 does not bring anything new to work, authors may consider removing it.
Referee is right, thus we have deleted the somewhat redundant figure 1 as indicated (this was also suggested by the referee No. 3).
Comment 2: Figure 2 letters are too small, please increase its font size.
Referee is right, this has been amended now
Comment 3: Editing of the English language and style are required.
We have extensively edited the manuscript.
Reviewer 3 Report
In this review, the authors firstly made a thorough description of the types, normalization attempts and isolation protocols of extracellular vesicles and biogenesis, compositions and secretion mechanisms of exosomes. Then, a detailed introduction to EV in the immune synapse formed by T lymphocytes upon TCR binding to antigen bound to APC was performed including exosome biogenesis, secretion mechanism and cytotoxic effects. Finally, CAR-T cell-derived EV about its potential therapeutic uses in cancer, the highlight of this review, was introduced. Overall, this review could generally provide advances in current knowledge of exosomes derived from CAR-T cells for readers and would benefit from a partial framework and content editing. To further make the review more readable and complete, some points must be concerned.
1. As to the length of this manuscript, the overview of extracellular vesicles/exosomes taking up a lot of space, especially in mechanisms of EV secretion, may confuse the readers interested in CAR-T cells-derived exosomes. From the relevance of the title and content, the length of each section should be rational and balanced after editing.
2. The role of EV from immune cells, including T lymphocytes, in anti-tumor immunity has been recently and exhaustively reviewed, and this review focus on potential therapeutic uses of CAR T cell-derived EV, so the title would be more accurate by replacing “T lymphocyte-derived exosomes and their applications in cancer therapy” with” CAR-T cell-derived exosomes and their applications in cancer therapy” that will benefit the corresponding audience.
3. There is a lack of summary of the current status of CAR-T therapy in cancer (solid tumors, hematologic tumors), although it currently shows some shortcomings.
4. In Line 297-301, the two sentences contain redundant information, which is a repetition of the same in different words.
5. It is known that nSMases inhibition decreases secretion of exosomes, but increases secretion of microvesicles (MVs) from the plasma membrane. The sentence in Line 359-360, “In contrast, nSMase blockade stimulates EV budding at the plasma membrane”, the “EV” should be replaced by “MVs” to avoid vagueness.
Author Response
In this review, the authors firstly made a thorough description of the types, normalization attempts and isolation protocols of extracellular vesicles and biogenesis, compositions and secretion mechanisms of exosomes. Then, a detailed introduction to EV in the immune synapse formed by T lymphocytes upon TCR binding to antigen bound to APC was performed including exosome biogenesis, secretion mechanism and cytotoxic effects. Finally, CAR-T cell-derived EV about its potential therapeutic uses in cancer, the highlight of this review, was introduced. Overall, this review could generally provide advances in current knowledge of exosomes derived from CAR-T cells for readers and would benefit from a partial framework and content editing. To further make the review more readable and complete, some points must be concerned.
- As to the length of this manuscript, the overview of extracellular vesicles/exosomes taking up a lot of space, especially in mechanisms of EV secretion, may confuse the readers interested in CAR-T cells-derived exosomes. From the relevance of the title and content, the length of each section should be rational and balanced after editing.
Referee is right. We feel that both the knowledge of the molecular bases involved in EV (exosomes and shedding vesicles) generation and secretion in T lymphocytes, and the description of EV molecular composition (presence of proapoptotic molecules, TCR and CAR, etc.), are crucial to understand how CAR T cells produce antigen-specific, bioactive EV, and how this can be tuned to improve EV-based therapy. That is the reason we dedicate several pages to this important introductory issue. In addition, since the exact nature of the EV (exosomes and/or shedding vesicles) involved in CAR T cells function is unclear (Table 1), we develop a thorough description of the types, normalization attempts and isolation protocols of extracellular vesicles. Otherwise the CAR-T cell-derived exosomes section would not be properly introduced and table 1 would be puzzling for a potential reader. In addition, to satisfy this referee’s point, we have balanced the introductory sections by deleting the somewhat redundant figure 1 (also suggested by the referee No. 2) and some spare sentences and including new sentences in Sections 6 and 7.
In addition, to further balance the manuscript, we have now introduced, out of the introductory pages, new sentences regarding NK-derived exosomes and CAR T cell memory cells, as requested by Referee 1. The title has been also changed (see next referee’s point 2).
- The role of EV from immune cells, including T lymphocytes, in anti-tumor immunity has been recently and exhaustively reviewed, and this review focus on potential therapeutic uses of CAR T cell-derived EV, so the title would be more accurate by replacing “T lymphocyte-derived exosomes and their applications in cancer therapy” with” CAR-T cell-derived exosomes and their applications in cancer therapy” that will benefit the corresponding audience.
Referee is right, thus we have changed the title. We choose “T lymphocyte-derived extracellular vesicles” (and not only exosomes) since we deal with the molecular bases involved in EV (exosomes and shedding vesicles) generation in T lymphocytes , and “CAR-T cell-derived extracellular vesicles and their applications in cancer therapy”, as indicated. In addition, since the exact nature of the EV (exosomes and/or shedding vesicles) involved in CAR T cells function is still unclear (see Table 1) we have used the inclusive term “extracellular vesicles” in the title, for coherence.
- There is a lack of summary of the current status of CAR-T therapy in cancer (solid tumors, hematologic tumors), although it currently shows some shortcomings.
Referee is right. We have included a new sentence in Sections 6 and 7 to deal with this important point.
- In Line 297-301, the two sentences contain redundant information, which is a repetition of the same in different words.
Referee is right, thus we have deleted the second, duplicated sentence.
- It is known that nSMases inhibition decreases secretion of exosomes, but increases secretion of microvesicles (MVs) from the plasma membrane. The sentence in Line 359-360, “In contrast, nSMase blockade stimulates EV budding at the plasma membrane”, the “EV” should be replaced by “MVs” to avoid vagueness.
Referee is correct. We have amended this sentence, as requested.
Round 2
Reviewer 1 Report
Overall, the authors have responded to my comments.
Reviewer 3 Report
To balance the manuscript, the author deleted the somewhat redundant figure 1 and some spare sentences and introduced NK-derived exosomes and CAR-T cell memory cells also edited the title. I find that the structure and contents of this review are very much improved. Overall, this revised review could generally provide advances in current knowledge of EVs derived from T cells and CAR-T cells for readers. As a result, I have no problem recommending it for publication.